# FSHD Therapeutic Strategies: What Will It Take to Get to Clinic?

**DOI:** 10.3390/jpm12060865

**Published:** 2022-05-25

**Authors:** Charis L. Himeda, Peter L. Jones

**Affiliations:** The Department of Pharmacology, University of Nevada, Reno School of Medicine, Reno, NV 89557, USA; chimeda@med.unr.edu

**Keywords:** facioscapulohumeral muscular dystrophy, FSHD, DUX4, skeletal muscle, muscular dystrophy, gene therapy, AAV, CRISPR, antisense, therapeutics

## Abstract

Facioscapulohumeral muscular dystrophy (FSHD) is arguably one of the most challenging genetic diseases to understand and treat. The disease is caused by epigenetic dysregulation of a macrosatellite repeat, either by contraction of the repeat or by mutations in silencing proteins. Both cases lead to chromatin relaxation and, in the context of a permissive allele, pathogenic misexpression of *DUX4* in skeletal muscle. The complex nature of the locus and the fact that FSHD is a toxic, gain-of-function disease present unique challenges for the design of therapeutic strategies. There are three major DUX4-targeting avenues of therapy for FSHD: small molecules, oligonucleotide therapeutics, and CRISPR-based approaches. Here, we evaluate the preclinical progress of each avenue, and discuss efforts being made to overcome major hurdles to translation.

## 1. Introduction

Facioscapulohumeral muscular dystrophy (FSHD) is one of the most prevalent myopathies that afflict males and females of all ages [1,2,3,4]. With the onset of clinical weakness typically appearing in the second or third decade of life and ~20% of patients ultimately using a wheelchair, the personal, social, and economic costs of this disease are enormous [5,6]. Although progress is being made, there are no cures or ameliorative treatments for FSHD, so an effective therapy is critically needed.

FSHD is caused by genetic mutations leading to epigenetic dysregulation of the D4Z4 macrosatellite repeat array at chromosome 4q35. FSHD1 is caused by contraction of the repeat, while FSHD2 is caused by mutations in silencing proteins (reviewed in [7]). Both cases lead to chromatin relaxation and, in the context of a permissive allele, misexpression of the *DUX4* gene in skeletal muscle. *DUX4* encodes a pioneer transcription factor that activates a program of gene expression during early human development [8,9,10], after which its expression is silenced in most somatic cells. When misexpressed in FSHD skeletal muscle, the DUX4 transcriptional program ultimately leads to myofiber degeneration with variable histopathologic features. Thus, in sharp contrast to most other genetic myopathies, which are caused by mutations in genes encoding muscle structural, regulatory, or signaling components, leading to a loss of function that needs to be fixed or replaced, FSHD is a toxic gain-of-function disease [11,12,13,14,15]. This necessitates the design of very different therapeutic strategies and, while avoiding some translational difficulties, introduces others.

While the most direct approach to therapy is eliminating DUX4 expression (Figure 1), both the gene and the D4Z4 repeat that encodes it present unique therapeutic challenges. Although highly similar D4Z4 arrays are found at multiple loci in the genome [16], *DUX4* is only stably expressed from the distal-most repeat unit on a disease-permissive allele [17,18,19,20] due to the presence of a polyadenylation signal (PAS) in an exon distal to the array (Figure 2) [20,21]. Additionally, while all FSHD myocytes are epigenetically poised to express *DUX4*, the transcript and protein are only expressed rarely, in stochastic bursts [21,22,23,24], consistent with the sporadic muscle involvement seen in FSHD patients. Moreover, while other mammals contain functional orthologs, the D4Z4 array and intact *DUX4* gene are not well conserved outside of old-world primates and the Afrotherian lineage, and no natural animal models exist [25,26,27,28].

The creation of FSHD-specific therapeutics has been understandably hampered by these challenges, and in the interim, easier avenues have been explored, including compensatory treatments to improve muscle strength or health [29,30,31,32,33,34,35,36] (ClinicalTrials.gov Identifiers: NCT02927080, NCT02603562), immunomodulation (ClinicalTrials.gov Identifier: NCT02579239), and most recently, the repurposed FDA-approved drug losmapimod [37] (ClinicalTrials.gov Identifiers: NCT04003974, NCT04264442). Thus far, these treatments have either failed or yielded underwhelming results in clinical trials. This is not unprecedented; many treatments across numerous indications have been quite successful in preclinical studies only to fail during translation. Events in the field of myotonic dystrophy have underscored the importance of not abandoning alternative avenues of therapy even in the wake of a very promising treatment [38]. Fortunately, DUX4 misexpression offers numerous candidates for therapy, and the field has benefited from rapid advances in both targeting modalities and modes of delivery. Thus, there are now many FSHD-specific therapeutics in the developmental pipeline (Figure 1 and Figure 2).

The bulk of work exploring FSHD therapeutic avenues has been recently and extensively reviewed elsewhere [39,40,41]. In this review, we evaluate the major avenues of therapy for FSHD in terms of their translational progress. We compare the advantages and disadvantages of each approach and take a hard look at the hurdles that must be overcome for each to reach the clinic.

## 2. Small-Molecule Drugs

These therapeutics are chemical compounds with a molecular weight in the range of 0.1–1 kDa that can target extracellular components or easily enter cells to affect intracellular molecules.

### 2.1. Progress in the Field

The discovery that FSHD pathogenesis is caused by misexpression of *DUX4* has spurred the identification of numerous potential therapeutic targets, both upstream and downstream of the pathogenic gene. However, FSHD creates complications not present in traditional drug screening. To begin with, DUX4 is only expressed in a few primary human tissues, and differentiated FSHD skeletal myocytes—the therapeutic target in vivo—are typically used for identifying repressors of DUX4 expression. In addition to the relative difficulties of growing and differentiating myoblasts, FSHD cultures typically exhibit extremely low and highly variable levels of DUX4 mRNA and protein. Thus, one must look for a decrease in levels of a transcription factor that is already expressed at very low levels, which is not an ideal situation for high-throughput drug screening. To address these issues, many academic and corporate researchers identify lead compounds through indirect expression screens using DUX4 reporter constructs as opposed to directly measuring *DUX4* transcript levels. This screening method is limited by the content of the chemical libraries screened, dosing, and modes of action. Highlighting these issues, despite the clear overlap in libraries, two published screens with similar approaches identified different molecules, targets, and pathways for *DUX4* inhibition, even to the exclusion of other targets [42,43], which is cause for concern. Regardless, screens for compounds that repress *DUX4* expression yielded bromodomain and extra-terminal (BET) inhibitors, agonists of the β2 adrenergic receptor [42,43], phosphodiesterase (PDE) inhibitors [43], p38 inhibitors [44,45], and Wnt agonists [46].

In contrast to chemical library screening, a candidate gene knockdown approach identified key regulators of *DUX4* expression (ASH1L, BRD2, KDM4C, and SMARCA5) that would be prime therapeutic targets for small-molecule inhibition [47]. Other studies identified the NuRD and CAF-1 complexes [48], PARP1 [49], and SMCHD1 [50] as endogenous D4Z4 repressors. Boosting the expression or activity of such factors, potentially through small molecules, represents an alternative avenue for therapy. In a different approach, a genome-wide CRISPR-Cas9 screen identified DUX4 pathways that could potentially be targeted for therapy [51]. While these and other targets downstream of DUX4 activity have been identified [51,52,53,54], it is not clear which, if any, of these downstream pathways are causal for pathology, or if successful targeting of any would provide therapeutic benefit. Additionally, many downstream factors are part of ubiquitous and robust cellular pathways (e.g., p300) [54]; thus, drugs targeting them are likely to produce a host of unwanted effects.

To date, only one putative small-molecule inhibitor of DUX4 expression has reached clinical trials in the US and Europe. The p38 inhibitor losmapimod emerged as a repressor of DUX4 expression in two independent screens [44,45]. Subsequently, Fulcrum Therapeutics licensed losmapimod, originally developed for the treatment of cardiovascular disease, from GlaxoSmithKline. Following early trials ([37]; ClinicalTrials.gov Identifier: NCT04003974), losmapimod is currently scheduled to enter a Phase III clinical trial for FSHD in Q2 of 2022 (ClinicalTrials.gov Identifier: NCT04264442), but the results thus far are not encouraging. Data reported in June 2021 from the Phase IIb trial (ClinicalTrials.gov Identifier: NCT04003974) showed only nominal benefit at 48 weeks, and the primary endpoint (reduction of DUX4 target gene expression) was not met. This indicates that any short-term functional improvements were DUX4-independent and, in the absence of *DUX4* repression to halt new muscle pathology, casts doubt on the likelihood of long-term benefit.

### 2.2. Advantages

While other forms of therapy are gaining ground, small molecules still hold significant advantages and represent the gold standard for treatment of most disorders [55]. Compared to biologics, small molecules are easier and cheaper to synthesize and generally more accessible to patients. They are generally taken orally, readily taken up by cells, stable, and not immunogenic. Additionally, the modality is well-established and inherently safer than any other.

### 2.3. Disadvantages

*Side-effects.* Small molecules can cause both off-target and on-target toxicity, with the latter being a particular concern for targets that are widely expressed and play key cellular roles. Losmapimod serves to illustrate this point, since it targets p38α, a key regulator of muscle biology [56,57,58]. Chronic therapeutic targeting of p38 for FSHD is quite different from protocols used for other indications; it requires hitting a narrow dosing window to repress *DUX4* expression without compromising important pathways such as muscle regeneration. Ultimately, the potential long-term negative effects of this dosing regimen are unknown. However, the D4Z4 repeat arrays are somewhat uniquely regulated in the human genome, and may offer more specific targets for therapy. Certain D4Z4/*DUX4* epigenetic regulators are not global regulators, which reduces the likelihood of harmful on-target effects. Additionally, even small changes in the levels of these factors can lead to striking decreases in *DUX4* expression [47]. This suggests that even incomplete target inhibition (i.e., very low doses) may be therapeutic, which should reduce the likelihood of both on- and off-target toxicity.

*Expensive over the long term.* A traditional drug treatment for FSHD will require chronic, lifelong administration.

### 2.4. Path to Clinic

Fulcrum Therapeutics, Myocea, Altay Therapeutics, and Facio Therapies, along with a number of academic labs, have reported active programs for small-molecule discovery and development in FSHD. Drugs for some candidate therapeutic targets (e.g., BET inhibitors, losmapimod) are already FDA-approved, while other targets are still in the initial phases of drug development. The repurposing of FDA-approved drugs as potential therapeutics for FSHD is attractive from a business standpoint. However, while these have been shown to be safe for indications such as cancer, they are not necessarily the safest or most efficacious treatments for a muscle disease. Drugs targeting ubiquitous cellular effectors are likely to have serious side effects, particularly during the long-term dosing required for FSHD. As mentioned above, even selective p38 inhibition via losmapimod may have harmful long-term effects in skeletal muscle, since p38 and its targets regulate many stages of myogenesis [56,57,58]. In addition, while targeting kinase pathways is a common clinical path, they are notoriously compensatory, which casts doubt on the likelihood of long-term inhibition [59].

The alternative route is to screen and optimize compounds targeting more specific therapeutic targets for which no drugs exist. While this approach is more costly and time-consuming, it enables the design of more specific drugs to better targets. The limited efficacy of losmapimod in the Fulcrum trial highlights the disadvantage of taking the short road to the clinic. In some cases, the long road may be the fastest route to a safe and effective therapy.

The following two sections deal with biologic therapeutics: compounds of biological origin, typically larger than small molecules, including peptides, proteins, nucleic acid-based molecules (oligonucleotides and gene therapies), and antibodies.

## 3. Oligonucleotide Therapeutics

### 3.1. Progress in the Field

FSHD is a dominant gain-of-function disease ideally suited for antisense or RNAi-based approaches. Over the last decade, various antisense approaches (Figure 2) have been tested against the *DUX4* gene and its pathogenic transcript, with considerable success in both in vitro and in vivo proof-of-principle studies.

The earliest antisense studies in the field focused on altering *DUX4* pre-mRNA processing. In healthy cells, *DUX4* can produce a short mRNA isoform that is translated into a nontoxic protein. In FSHD myocytes, there is a shift in mRNA splicing to generate the full-length, pathogenic *DUX4* isoform (*DUX4-fl*). Initial studies geared toward therapy showed that siRNAs and antisense oligonucleotides targeting the *DUX4* PAS and mRNA splice sites reduced levels of DUX4-fl mRNA and protein in FSHD myotubes [60,61]. When targeted to the *DUX4* promoter, siRNAs induced a DICER/AGO-dependent epigenetic transcriptional silencing [62]. Phosphorodiamidate morpholino oligomers (PMOs) targeting the *DUX4* PAS reduced levels of DUX4 and its targets in cultured FSHD myocytes [63,64] and following intramuscular delivery to a xenograft model [64]. LNA gapmers (against exon 3) and 2′-O-methoxyethyl (2′-MOE) gapmers (against the coding region) reduced DUX4 levels in immortalized FSHD myocytes and following intramuscular delivery to the *FLExDUX4* FSHD mouse model [65,66]. Importantly, in a recent study, PAS-targeting PMOs delivered systemically reduced DUX4 and its targets, ameliorated pathology, and improved muscle function in the *FLExDUX4* FSHD mouse model [67,68].

*DUX4*-targeting miRNAs were shown to reduce DUX4 mRNA and protein in a DUX4-overexpression mouse model [69,70], and U7 antisense DUX4 snRNAs reduced levels of DUX4 and its targets in cultured FSHD myocytes [71].

A different oligonucleotide approach utilizes DNA binding site decoys (DNA aptamers) to bind a transcription factor and block its ability to regulate target genes. Such decoys can bind recombinant DUX4-FL with high affinity [72], and others were demonstrated to block binding of DUX4 to its target genes following electroporation of DUX4 into wild-type mice [73].

### 3.2. Advantages


-ability to directly repress expression of DUX4, a difficult-to-drug transcription factor-relatively straightforward to design, produce, and screen-sequence-based targeting allows for precision/personalized medicine-treatment can be stopped if adverse events occur


### 3.3. Disadvantages

*Off-target effects/cytotoxicity.* All oligonucleotide drugs have the potential for sequence- and chemistry-dependent cytotoxicity and immunogenicity. Sequences that function efficiently in cell culture may have inherently bad toxicology profiles in non-human primates (NHPs) and humans. Oligonucleotide therapeutics that use RNAi are notoriously cytotoxic. miRNAs are inherently less specific than ASOs because they bind via incomplete base-pairing, increasing the risk of off-target effects. siRNAs are double-stranded, which increases the risk of an innate immune response.

*Dysregulation of endogenous RNA-processing pathways.* All oligonucleotide drugs utilize endogenous cellular enzymes to mediate their effects. siRNAs, shRNAs, and miRNAs depend on RNAi, thus competing with normal substrates of the RISC complex and sometimes oversaturating the capabilities of this pathway [74]. Additionally, some ASOs that depend on RNase H for target degradation can induce “ASO tolerance” via increased expression of the target pre-mRNA [75].

*Inefficient uptake into muscle cells.* In general, oligonucleotide drugs suffer from poor delivery to tissues other than liver.

*Requirement for chronic, lifelong administration.* This requirement will likely exacerbate any toxicity caused by the treatment.

*Approved oligonucleotide drugs will likely be extremely expensive over a lifetime.* To date, there is no publicly available data on durability and dosing for these compounds in FSHD.

### 3.4. Path to Clinic

Oligonucleotide drugs for multiple diverse disorders, including Duchenne Muscular Dystrophy (DMD), have recently been approved by the FDA [76], demonstrating that this technology has reached clinical maturity for some indications. For FSHD, it is encouraging that a number of biotech companies (Avidity Biosciences, Dyne Therapeutics, Arrowhead Pharmaceuticals, miRecule, and Armatusbio) are actively developing DUX4-targeting oligonucleotide therapeutics for FSHD.

*Delivery challenges must be addressed.* Fortunately, there is a major push to develop technologies improving the drug-like properties of nucleic acid therapeutics for multiple indications. These include chemical modifications, backbone modifications, conjugation to a carrier that facilitates uptake by the target tissue(s), and nanocarriers [76]. For some indications, nucleic acid oligomers (e.g., mRNA vaccines) are now in use, thanks in large part to efficient delivery by lipid nanoparticles (LNPs). Although targeting of LNPs to body-wide skeletal muscles poses different challenges and has yet to be optimized, cell-penetrating peptide conjugate oligonucleotides may improve delivery to muscle. Avidity Biosciences and Dyne Therapeutics, two companies with active FSHD programs, have reported the development of antibody conjugates that improve muscle uptake. Oligonucleotide drugs can also be delivered by gene therapy vectors (see below).

*Specificity must be rigorously assessed for every candidate oligonucleotide drug at efficacious doses.* Thus far, only two studies have reported RNA-seq data following treatment [64,66]. (Lim et al. focused only on the correction of FSHD signature gene expression, whereas Chen et al. reported no significant off-target effects of their PMO treatment on the muscle transcriptome.) Additionally, it is important to assess off-target effects in muscle cells that are not expressing DUX4, since these will be the overwhelming majority in a patient subjected to treatment, and the specificity of an oligonucleotide drug may decrease in the absence of a perfect target.


*Potential immunogenicity/toxicity of any carriers/conjugates must be assessed at efficacious doses.*


*Long-term efficacy must be assessed.* Thus far, the longest time point assessed in any FSHD study is following 4 weeks of treatment in a mouse [68]. The critical question is: can an oligonucleotide drug be administered systemically and chronically at a dose that provides functional benefit without toxic effects? Eteplirsen and golodirsen, FDA-approved exon-skipping drugs for DMD, have shown little clinical benefit, and it can be argued that the situation in FSHD is more challenging to treat. Treatment of DMD requires production of functional dystrophin at levels far below those found in healthy individuals and in only a fraction of myofibers, whereas treatment of FSHD may well require much higher doses to achieve DUX4 repression, which will likely be required in a majority of myofibers, since every muscle nucleus is poised to express DUX4 at any given time.

## 4. Gene Modification

### 4.1. Progress in the Field

Over the past few years, several groups have attempted CRISPR modulation of the FSHD locus, either by Cas9-mediated gene editing (CRISPRe) or dCas9-mediated transcriptional inhibition (CRISPRi) (Figure 2 and Figure 3). Recently, the Dumonceaux lab demonstrated that editing the *DUX4* exon 3 PAS (by Cas9 or TALENs) is highly inefficient [77]. Even in successfully edited clones, pathogenic *DUX4* levels are reduced, but not silenced, likely due to compensation by a cryptic PAS; the authors concluded that this is not a viable therapeutic strategy for FSHD [77]. By contrast, another group attempting a similar approach reported that CRISPRe of the exon 3 PAS reduced D4Z4 transcripts and expression of DUX4 target genes in immortalized FSHD1 myoblasts [78]. RNA-programmable base editing targeting the *DUX4* PAS (18% efficiency) reduced levels of *DUX4* and its target genes in both FSHD1 and FSHD2 myoblasts, and a cursory assessment of specificity was encouraging [79]. Finally, CRISPRe correction of an *SMCHD1* intronic variant restored *SMCHD1* expression to healthy levels and reduced *DUX4* expression [80].

Our group has taken a CRISPRi approach to mitigate the concerns related to genome editing, and because CRISPRi is ideally suited to correcting a disease caused by the pathogenic misexpression of a powerful transcription factor [81,82,83] (Figure 2 and Figure 3). Targeting the enzymatically inactive dSaCas9 fused to one of several epigenetic repressor domains to the *DUX4* promoter/exon 1 increases chromatin repression at the disease locus, specifically repressing *DUX4* and its target genes in FSHD myocytes and in a mouse model of the disease [83]. We have recently re-engineered this platform to a single-vector system in which all therapeutic components can be packaged into recombinant AAV (rAAV) vectors for gene therapy.

### 4.2. Advantages


-Both CRISPRe and CRISPRi have the potential to achieve long-term or permanent correction of all forms of FSHD with a single treatment.-In principle, CRISPRi is ideally suited to a dominant, gain-of-function disease such as FSHD. Since it does not involve cutting the genome, CRISPRi may also be safer than CRISPRe.-In the long run, CRISPR therapies (and other gene therapy approaches) may be the most economical means of treatment, since they are delivered in a single dose that is meant to function over a lifetime.


### 4.3. Disadvantages

*Challenges related to rAAV-mediated gene therapy* (see below).

*Immunogenicity related to CRISPR components* [84]. In the one in vivo clinical trial of CRISPR gene editing to date, it is encouraging that no significant safety concerns were reported in the six patients at 28 days post-treatment [85]. However, several studies have demonstrated that Cas9 evokes a host cellular and humoral immune response, and a large proportion of the population has pre-existing immunity to Cas9 [84]. Thus, developing strategies to avoid a potential Cas9 immune response is critical. These include: (1) removing immunogenic Cas9 epitopes (immunodominant epitopes have been removed from SpCas9 while preserving its cutting efficacy) [86], (2) interfering with processing and presentation of Cas9 antigens, (3) using orthologs from nonpathogenic bacteria, (4) inducing immune tolerance, and (5) transient immunosuppression [84]. miRNA-based detargeting has also emerged as a way to specifically repress expression of immunogenic transgenes in antigen-presenting cells by incorporating binding sites for macrophage-specific miRNAs into rAAV vectors [87,88].

*Off-target effects.* These are a bigger concern for CRISPRe than for CRISPRi approaches, since off-target cutting is likely to be more detrimental than off-target repression.

*Genomic instability.* This is only a concern for CRISPRe approaches, which involve genomic cutting.

### 4.4. Path to Clinic

At least one company (EpiSwitch Rx) has an active CRISPR program for FSHD.
-For CRISPRe, rigorous tests of specificity are required. Rapid advances in specific, programmable base-editing (e.g., prime-editing) [89] should greatly improve the safety of CRISPRe for FSHD. However, current base editors are much too large to be accommodated in a single-vector system for rAAV-mediated delivery [89]. These need to be better characterized and minimized to be therapeutically relevant.-Stable repression mediated by CRISPRi remains to be demonstrated and is absolutely necessary to achieve, since current rAAV vectors for gene therapy can only be administered once. In brief, either the CRISPRi treatment must elicit a stable epigenetic change or the rAAV episome must persist indefinitely in treated fibers (discussed in more detail below).-Both types of CRISPR approaches must be tested more rigorously in appropriate in vivo models, including a large animal model (see below). For CRISPRi, a human xenograft model [90,91] is ideal for assessing the persistence of epigenetic changes at the disease locus, as xenografted mice contain a full D4Z4 array from an FSHD patient.

## 5. Needs for Therapeutics Delivered by Gene Therapy (CRISPR Strategies, RNAi)

### 5.1. Delivery to Body-Wide Skeletal Muscles

Currently, rAAV vectors are the only practical way to deliver Cas9/dCas9 components to skeletal muscles throughout the body. Several AAV serotypes with high muscle tropism have been widely used in preclinical studies and in clinical trials for DMD and X-linked myotubular myopathy (XLMTM) (clinicaltrials.gov identifiers: NCT03362502, NCT03368742, NCT03375164, and NCT03769116). Despite the utility of these vectors, the need remains to increase the efficiency of delivery, lower the high cost of therapy, and reduce the liver and immune toxicity associated with high viral doses. For these reasons, rAAV-mediated gene therapies should use therapeutic cassettes contained within single vectors [92].

Recently, two groups have reported engineered capsid variants (“AAVMYO” and “MyoAAV”) with greatly enhanced tropism for striated muscles combined with detargeting of the liver [93,94]. These capsids demonstrate greater efficiency and selectivity of muscle targeting than those currently being used in clinical trials and in comparable preclinical studies. The MyoAAV class was shown to be highly muscle-tropic in both mice and NHPs, with therapeutic efficacy demonstrated following systemic administration of a low dose in multiple disease models [94]. This ability to lower the effective vector dose by even several-fold represents a major safety and efficacy milestone for gene therapies of muscle diseases.

### 5.2. Overcoming rAAV-Related Limitations

Gene therapies using rAAV vectors have been used safely and successfully in many clinical trials, and three (Glybera, Luxturna, and Zolgensma) have been approved. Nonetheless, use of these delivery vectors has several limitations.

*Neutralizing antibodies to the capsid prevent re-administration.* Although efforts are being made to reduce the immunogenicity of rAAV vectors [95,96], gene delivery by rAAV is currently a single-dose injection; thus, stability of the treatment is key. For FSHD, this means that either the rAAV episome must persist indefinitely in infected fibers, or that the treatment itself must elicit a stable change (e.g., epigenetic repression at the disease locus). Whether the transduced vector will eventually be diluted past the point of therapeutic efficacy is unknown, but rAAV episomes have shown remarkable persistence in skeletal muscle, a largely quiescent tissue, assembling into active chromatin and mediating transgene expression for several years in the muscles of NHPs [97]. In clinical trials for DMD, micro-dystrophin delivered by rAAVrh74 has persisted for over 3 years [98], and for spinal muscular atrophy, survival motor neuron delivered by AAV9 has proven efficacious for over 5 years [99].

*Pre-existing immunity to AAV renders a certain percentage of the population untreatable* [89]. Since neutralizing antibodies can cross-react with multiple AAV serotypes, switching serotypes is not a viable strategy to avoid neutralization of rAAV vectors [100]. More realistic strategies to circumvent this include: (1) transient or long-term immunosuppression [101], (2) plasmapheresis to deplete AAV-specific neutralizing antibodies [102], (3) treatment with IgG-cleaving endopeptidases to decrease anti-AAV antibody titers [103], and (4) chemical modification [104], capsid engineering [105], or use of empty capsid decoys [106] to protect rAAV vectors from neutralization. The capsid can also trigger clearance of infected cells by cytotoxic T lymphocytes [107], an event that can be mitigated by Treg cells [108]; thus, autologous Treg cells may prove to be a useful adjuvant to in vivo gene therapies [109]. For most of these strategies, effectiveness in clinical settings remains to be tested. Additionally, benefits may not be straightforward, as with the use of empty capsids [110].

*High doses cause liver toxicity.* Recent clinical trials for DMD and XLMTM (clinicaltrials.gov identifiers: NCT03368742 and NCT03199469) were put on hold by the FDA due to severe adverse events associated with liver toxicity [111]. Although the hold on the DMD trial has since been lifted, this highlights the urgent need to detarget the liver and lower efficacious doses, which is currently being met [93,94].

*Potential genotoxicity caused by genomic integration.* Predominantly episomal, rAAV vectors have been found to integrate in the host genome at very low frequencies. It is not known whether long-term persistence of rAAV vectors in largely quiescent tissues such as skeletal muscle might eventually result in genotoxicity due to vector genome integration; however, the evidence to this point has led to a general consensus within the gene therapy field that such events are extremely unlikely [112].

*High cost of production.* Approved gene therapies using rAAV vectors are astronomically expensive. Priced at USD 1.2 million, Glybera has already been withdrawn from the market after treatment of just a single patient. At USD 2.1 million, Zolgensma is now the most expensive drug in the world. While the economics of an effective single-dose gene therapy vs. a chronic, lifelong treatment are beyond the scope of this review, high manufacturing costs can be lowered by lowering the effective dose. This can be accomplished by: (1) reducing packaging of partial/truncated genomes and empty capsids that co-purify with the therapeutic vector [110,113,114,115], (2) reducing immunogenicity [95,96], and (3) detargeting liver while increasing delivery to skeletal muscle [93,94].

## 6. Needs for All DUX4-Targeted Therapeutics

### 6.1. A Large Animal Model of FSHD for Preclinical Testing of Therapeutics

It has long been established that mouse models do not accurately reflect important features of human disorders. In addition, the delivery, dosing, specificity, efficacy, and durability of treatments are highly variable between a mouse (~25–35 g) and an adult human (~45–135 kg). As a result of these differences, many treatments have been quite successful in mouse models only to fail during clinical trials. Testing candidate therapeutics in a large animal model with closer genetic, anatomical, physiological, and metabolic similarities to humans should go a long way toward solving this problem.

However, rising ethical concerns have cast disfavor on the use of NHPs and dogs in biomedical research, and when used for testing therapeutics, these studies are often underpowered due to their high costs. Minipigs have come to fill this gap and are becoming established as the large animal model of choice for translational research [116,117,118]. They share many similarities to humans in terms of size, physiology, anatomy, metabolic profile, and lifespan, while offering lower operating costs and the ability to perform properly powered experiments to determine both efficacy and toxicology [119,120,121]. To fill this need, the first large animal model of FSHD, a Göttingen minipig based on conditional expression of DUX4-fl in skeletal muscles [122], was generated and is currently being validated with respect to FSHD gene expression signature and inducible myopathy. Most FSHD therapeutics under development require only a large animal model with consistent, inducible expression of DUX4-fl and its target genes in order to test target engagement, molecular efficacy, dosing, and delivery. Thus, while a phenotypic large animal model would be ideal, any model that produces mosaic DUX4 expression in skeletal muscle cells will still be extremely useful for the therapies described above.

### 6.2. DUX4-Responsive Circulating Biomarkers of Disease Progression

One of the greatest unmet needs in the FSHD field is a robust and reliable circulating biomarker for pathogenic gene expression or disease progression that can be used to readily monitor the efficacy of a treatment. Compared with muscle biomarkers, serum biomarkers are minimally invasive, inexpensive, and ideal for monitoring drug responsiveness over time. For drugs targeting *DUX4* expression, a DUX4-responsive circulating biomarker is needed, since there is no consensus on which genes downstream of DUX4 reflect pathological changes in FSHD muscle.

Several studies have identified proteins, mRNAs, and miRNAs that are differentially expressed in the serum of FSHD patients vs. healthy controls [123,124,125,126,127]. Among these, miR-206 was shown to be DUX4-responsive, and thus, a strong candidate for a biomarker to assess DUX4-targeting treatments [127]. Levels of IL-6, a cytokine and myokine that is elevated in several neuromuscular diseases, were found to correlate reliably with clinical severity, making it a good potential biomarker for tracking disease progression [128]. All candidate biomarkers to date require independent validation in large patient cohorts, as well as longitudinal evaluation.

The current state of clinical outcome measures, including imaging biomarkers, has been reviewed in detail [129]. These authors call for the use of circulating biomarkers of active muscle disease in combination with whole-body MRI and comprehensive quantitative assessments of muscle function [129], all of which are undergoing development and optimization. In addition, standardization of patient procedures is absolutely critical when interpreting outcomes. Standard operating procedures regarding fasting, exercise, ambient temperature, etc. must be established and rigorously adhered to.

The development of biomarkers and clinical outcome measures is particularly challenging for FSHD, considering the heterogeneity of muscle involvement and the highly variable and unpredictable course of disease progression. The usefulness of any outcome measures depends on the establishment of proper enrollment criteria, which will vary depending on the treatment, and proper patient stratification. The latter cannot depend exclusively on repeat size, since this does not correlate with clinical severity [130,131,132,133]. Clinical severity, type of FSHD, presence of modifiers, degree of D4Z4 hypomethylation, and location of muscle involvement must all be considered during stratification.

## 7. DUX4-Independent Approaches

Incapacitating DUX4 and halting pathology are the primary goals of most therapeutic programs for FSHD. However, certain DUX4-independent interventions may prove useful as part of a combination therapy or adjuvant treatment.

Treatment with oral antioxidants yielded minimal benefit in a recent FSHD trial [35], but outcomes might be improved with mitochondria-targeted antioxidants [134] or by targeting multiple common pathways of mitochondrial dysfunction [135]. In this regard, aerobic exercise improved mitochondrial function and physical performance in DM1 patients [136], and might be helpful to FSHD patients, as well. Creatine supplementation has also improved muscle strength across multiple muscular dystrophies [137]. Regarding immune suppression, anti-IL-6 receptor monoclonal antibodies tocilizumab and sarilumab, approved for other indications [138], might be tested in FSHD, although on-target effects are likely to be an issue. While the immunomodulator ATYR1940 was unsuccessful in a clinical trial for FSHD, it is feasible that an anti-inflammatory treatment in combination with a DUX4-targeting therapy might be effective at halting disease progression and restoring a healthy muscle environment.

Despite failure in a number of clinical trials, the book is not entirely closed on myostatin inhibition as a means of increasing muscle strength under the right conditions. The clear discrepancy between clinical trial results and effectiveness in mouse models highlights: (1) the difference between mouse and human, and the need for large animal models of disease, and (2) the need for properly stratified and controlled clinical trials. Often, a single trial does not provide a straightforward answer, as myostatin inhibition failed for different reasons in different trials [139]. Importantly, circulating baseline levels of myostatin are higher in FSHD than in DMD, and the ACE-083 trial demonstrated significant muscle hypertrophy (without an increase in strength) in treated FSHD patients [36]. Thus, it is conceivable that in combination with a DUX4-targeting therapy, myostatin inhibition might increase muscle function as well as size in FSHD.

The need for well-planned clinical trials becomes more pressing as a community grows increasingly desperate for a treatment, and is willing to grasp at untested straws. Two of these—stem cell therapy and exosome treatments—may be useful for providing local immune suppression, but their safety, efficacy, and durability are still untested for FSHD, even in preclinical models.

The potential benefits of stem cell implantation (generally used to repair or replace muscle in an otherwise healthy environment) are by no means straightforward in a dystrophic environment such as FSHD muscle. As the myostatin trials demonstrate, effects in healthy muscle are not necessarily predictive of effects in dystrophic muscle. Nonetheless, the efficacy of stem cell therapy has been well documented for other indications, and is worth testing in carefully planned, well-controlled, and properly powered clinical trials for FSHD. By contrast, the use of exosomes to carry cytokines or growth factors is promising, but these vesicles are just beginning to be characterized. Their use in a clinical setting is untested and unapproved, yet unregulated clinics have sprouted up in many countries, including the US, and are flourishing under the notion that the potential benefits are worth the risks. This is a serious concern because the risks are not trivial. Beyond the risks associated with the unregulated use of biologics from questionable sources, treatments that foster immune tolerance may only provide short-term benefit, while enabling the growth of early-stage tumors. Additionally, any treatment that stimulates muscle regeneration (as stem cell therapy is supposed to do) may deplete muscle satellite cells, thus hastening dystrophic progression.

None of these treatments will correct the underlying cause of FSHD, but one or more may ultimately prove useful, likely in combination with a DUX4-targeting therapy. When considering a general ameliorative treatment for FSHD, one must ask: (1) what are the likely benefits and risks in a dystrophic environment? (2) what are the probable effects in cells that are either expressing or poised to express DUX4? and (3) what are the best models for testing? While animal models are not always predictive of the response in humans, they have been indispensable for revealing the effects of perturbing conserved pathways and mechanisms.

## 8. Conclusions and Perspective

Despite the complexities of the disease and the locus, the FSHD field has made considerable progress since the pathogenic role of DUX4 was first identified. The fact that multiple modalities targeting DUX4 expression and activity are moving through the therapeutic pipeline is promising for the prospect of an effective treatment in the near future. With technology advancing rapidly on multiple fronts, it is hard to predict which avenue will be first to reach the clinic. Regardless, the first drug to market may not ultimately be the best; thus, it is important to continue developing other viable candidates while the long-term efficacy of any approved treatment is still in question.

Many tools have been developed and tested in proof-of-principle studies that do not always make the difficult leap to clinical relevance. Indeed, preclinical data must be especially robust and reliable for a rare disease where the number of participants is limited and clinical trial failure is highly damaging to the quest for any effective therapy [140]. Currently, there is a pressing need for studies that address the important translational questions and obstacles associated with each of the therapeutic avenues discussed here. Antisense approaches must demonstrate efficient uptake into muscle cells and both efficacy and lack of toxicity following long-term treatment. Similarly, CRISPR approaches must demonstrate long-term efficacy and safety, while mitigating Cas9 and AAV immunogenicity. One question that concerns all therapeutic approaches is: what level of DUX4 inhibition is required for functional benefit? While DUX4 is only expressed in rare myonuclei at any given time, every myonucleus in an FSHD patient is epigenetically poised to express DUX4. Thus, both antisense and CRISPR approaches will likely need to target a majority of myonuclei in a majority of muscle fibers throughout the body. Fortunately, all available data indicate that any reduction in DUX4 levels is likely to be therapeutic. In addition to the fact that asymptomatic individuals still express detectable levels of *DUX4* [22,141], small increases have catastrophic effects in FSHD mouse models [122,142]. Thus, there is good reason to believe that even small decreases in DUX4 expression will be beneficial to patients.

FSHD is highly pleiotropic and the best treatment may prove to be a combinatorial or personalized approach that depends on disease type and severity. This may consist of a DUX4-targeted therapy in combination with a non-targeted approach, such as immune modulation or stem cell therapy. It is important to keep in mind that from a patient standpoint, simply halting disease progression does not constitute a real cure. To correct the primary defect in FSHD and also replace the loss of diseased muscle with healthy, functional muscle remains the ultimate goal. With continual progress in the many disciplines involved, such a prospect is no longer unfeasible to envision.

## Figures and Tables

**Figure 1 jpm-12-00865-f001:**
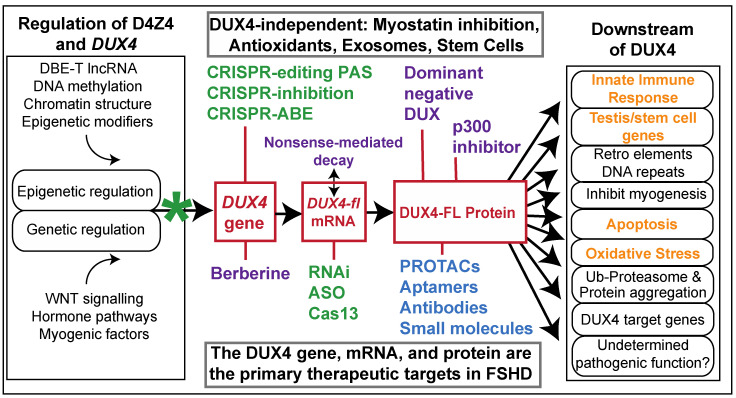
Incomplete silencing of the FSHD locus leads to misexpression of the pathogenic *DUX4-fl* transcript in skeletal muscles. DUX4-FL subsequently activates a program of embryonic gene expression, which triggers many aberrant downstream events. While some pathways downstream of DUX4 are amenable to targeting (orange), it is not clear which individual pathway, if any, is causal for pathology. Thus, serious efforts to develop specific FSHD therapeutics (green) have focused on restoring silencing to the locus or preventing expression of DUX4-fl mRNA or protein. Purple indicates DUX4-targeting strategies that are inherently promiscuous; asterisk indicates small molecules targeting *DUX4* epigenetic regulation or transcription. Blocking DUX4-FL activity (blue) is a theoretically viable avenue, although still in the very early stages of development.

**Figure 2 jpm-12-00865-f002:**
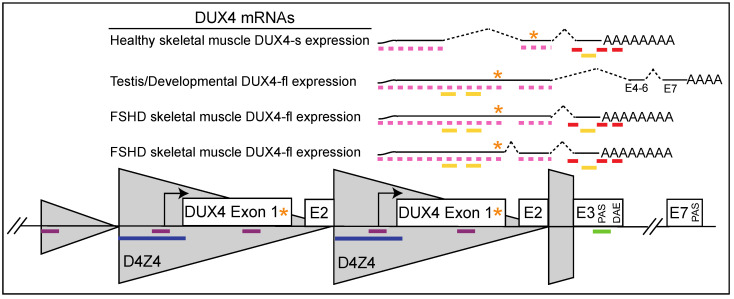
Therapeutic targets within the FSHD locus. Two D4Z4 repeats (grey) of a disease-permissive allele are depicted, along with distal sequences. Exons 1 and 2 are located within the repeat, and exon 3 is located in the distal sequence. Full-length *DUX4* transcripts (*DUX4-fl*) are expressed in cleavage-stage embryos and stabilized by a polyadenylation signal (PAS) in exon 7. Following this stage, expression of *DUX4-fl* is normally silenced in most somatic tissues, although the locus can produce unstable, nonpathogenic, short *DUX4* isoforms (not shown). In FSHD myocytes, incomplete somatic silencing at the locus mediates a switch from the production of these nonpathogenic isoforms to pathogenic *DUX4-fl* transcripts, which are stabilized by an exon 3 PAS within permissive haplotypes. Regions for therapeutic targeting are indicated by colored bars: antisense oligonucleotides (pink and red bars; the latter are reported sequences), microRNAs (yellow bars), CRISPR inhibition/epigenetic modulation (purple bars), CRISPR editing (green bar), and small interfering RNAs recruiting the Dicer/Argonaute silencing system (blue bar). The orange asterisks indicate DUX4-FL stop codon. Additional abbreviation: DAE, distal auxiliary element.

**Figure 3 jpm-12-00865-f003:**
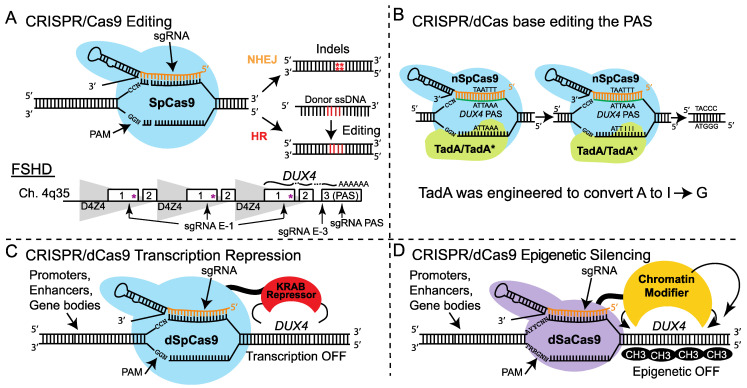
CRISPR gene modification approaches to FSHD. (**A**) The Cas9 nuclease cuts DNA at sites targeted by single-guide RNAs (sgRNAs). DNA repair by nonhomologous end joining (NHEJ) mediates the disruption of genomic sequences by insertions/deletions (indels), whereas homology-directed repair (HDR) mediates precise editing in the presence of a donor template. CRISPR editing of the D4Z4 repeat (*) necessitates cutting the genome in hundreds of unintended places, as these repeats exist in many copies on the noncontracted 4q chromosome and both 10q chromosomes, in addition to polymorphic D4Z4s at other loci. Thus, groups attempting CRISPR editing have focused on the polyadenylation signal (PAS) in *DUX4* exon 3, which provides a safer target. (**B**) RNA-programmable base editing utilizes a wild-type tRNA adenosine deaminase (TadA) and an engineered version of TadA (TadA*) fused as a dimer to a nicking version of Cas9 (nCas9). This enables adenine-to-guanine substitution without introducing double-stranded DNA breaks. (**C**,**D**) As an alternative strategy, enzymatically dead Cas9 (dCas9) is fused to the KRAB domain for transcription inhibition (panel **C**) or to chromatin-modifying proteins, which can act in a broader fashion across the locus (panel **D**). Fusing minimized versions of these regulators to dCas9 from *Staphylococcus aureus* allows packaging into rAAV vectors to silence pathogenic *DUX4* expression in FSHD muscle. Additional abbreviation: PAM, protospacer-adjacent motif.

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
