# Peer review of "FSHD Therapeutic Strategies: What Will It Take to Get to Clinic?"

_jpm, 2022, doi:10.3390/jpm12060865_

Round 1

Reviewer 1 Report

This review is a very comprehensive description of the therapeutic strategies and their path to clinic in FSHD. It is very well structured, clear and at the same time deeply deals with a complex topic, highlighting pro and cons of the different approaches. The only minor comment I could make is that in the paragraph "DUX4-independent approaches" the description of the inflammatory component, extensively studied both in vivo (by MRI) and on muscle biopsies (pathology, transcriptomic signature) and sometimes in combination [for example Hum Mol Genet 2019; 28(3):476–486, Mol Neurobiol 2018;55(4):2959-2966, PLoS One 2012; 7(6):e38779], is barely mentioned. This target could be particularly important in the context of combined/personalized therapies, as suggested by the authors. An unsuccessful clinical trial with a potential immunomodulatory drug (aTyr pharma, Resolaris) has been performed, too [http://dx.doi.org/10.1016/j.nmd.2016.06.293] . In the opinion of this reviewer, these topics are worth mentioning. 

Author Response

We thank the reviewer for the comment regarding immune approaches & the aTyr trial, and have incorporated this in the context of combined/personalized therapies.

Reviewer 2 Report

The review from Himeda and Jones is interesting, well-organized, and a useful summary of therapeutic strategy for the complex FSHD myopathy.

In the introduction the sentence 'When misexpressed in
FSHD skeletal muscle, the DUX4 program leads to accumulated muscle pathology' do not make sense. Please reformulate ....the DUX4 program leads to myofibers degeneration with variable histopathologic features  doi: 10.1002/mus.24621.

Figure 1 :  there is no explanation for the purple color

PMO PHosphorodiamidate morpholino oligomer : the full name is missing

Same for 2’ O methoxyethyl

I have some comments regarding the CRISPRi and CRISPRe.

Do not you think that patients would not need to integrate the CRISPR/dCas9 and SgRNA in their genome to have a permanent production and then correction (toxicity risk) ?

This especially in comparison to CRISPRe which indeed cut the DNA but could be used once.

There is no degradation of the CRISPRi complex over the time ?

CRISPRi complex would then have to be permanently present in the patient with more immune issues than CRISPRe ?

Author Response

We thank the reviewer for the comments, which helped to enhance and clarify the text.  We have made all of the suggested changes.  Regarding the comments about CRISPR: this is a good point, which we addressed in the AAV section, but some mention needed to be made in the CRISPR section as well, which we have included.